# Pertussis Vaccination Failure in the New Zealand Pediatric Population: Study Protocol

**DOI:** 10.3390/vaccines7030065

**Published:** 2019-07-16

**Authors:** Hannah Chisholm, Anna Howe, Emma Best, Helen Petousis-Harris

**Affiliations:** 1Department of General Practice and Primary Health Care, School of Population Health, Faculty of Medical and Health Sciences, University of Auckland, Auckland 1142, New Zealand; 2Department of Paediatrics, Child and Youth Health, School of Medicine, Faculty of Medical and Health Sciences, University of Auckland, Auckland 1142, New Zealand

**Keywords:** pertussis, vaccination failure, pediatric, study protocol

## Abstract

Pertussis vaccines have been effective at reducing pertussis-associated morbidity and mortality. However, they have a complex array of limitations, particularly associated with the duration of protection against clinical disease and imperfect immunity (carriage and transmission). Little is known about risk factors for pertussis vaccination failure. Understanding pertussis vaccination failure risk is most important in the paediatric population. This study aims to investigate risk factors for pertussis vaccination failure in (1) infants between birth and six weeks of age born to mothers who received pertussis booster vaccinations during pregnancy and (2) infants after the completion of the primary series (approximately five months old) to four years old. This will be achieved in a two-step process for each study group. Pertussis vaccination failure cases will first be described using a case series study design, relevant case characteristics will be sourced from six national administrative datasets. The case series study results will help select candidate risk factors (hypothesis generating step) to be tested in the retrospective cohort study (hypothesis testing step). Pattern analysis will be used to investigate risk factor patterns in the cohort study. The identification of higher risk groups enables targeting strategies, such as additional doses, to better prevent pertussis disease.

## 1. Introduction

Pertussis is a globally endemic and highly infectious disease that can cause respiratory, nutritional and neurological complications, and death [1]. Infants and young children have the highest risk of pertussis sequelae and death during the window of vulnerability (between birth and the first infant pertussis vaccination) [2]. Since the introduction of whole-cell pertussis vaccines (wP) in the 1940s, global pertussis-related morbidity and mortality have decreased substantially, particularly in high-income countries [3]. However, despite vaccination, an estimated 16 million pertussis cases and 195,000 child deaths occur globally every year, with the greatest burden in low and middle-income countries [4]. Furthermore, many high-income countries with high and stable vaccination rates, such as New Zealand, Australia and the United States, have increasing pertussis burden and regular epidemics are not being prevented [1].

Acellular pertussis vaccines (aP) are now used instead of wP vaccines in most high-income countries. In comparison to wP vaccines, aP vaccines are less reactogenic, yet have a lower effectiveness [5,6,7,8]. Estimates of aP vaccine effectiveness (VE) vary substantially with age, number of doses, outcome (hospitalisation/non-hospitalised notification/all cases), local epidemiology and health care service policies. For example, a New Zealand nested case-control study estimated a 93% VE against pertussis hospitalisation in the first year of life for a three-dose primary pertussis vaccination series (DTaP, Infanrix^®^-Hexa, GlaxoSmithKline, Rixensart, Belgium) [9]. However, an Australian case-control study estimated 85% VE against pertussis hospitalisation in the first year of life for a three-dose primary pertussis vaccination series (DTaP combination vaccines, GlaxoSmithKline, Rixensart, Belgium) [10]. Both studies used national-level registry data but the lower VE in the Australian study may in part be due to the high polymerase chain reaction (PCR) testing for *Bordetella pertussis* in Australian health services compared with New Zealand [9]. As aggressive PCR testing for pertussis is not commonplace, it is likely that aP VE is lower than previously reported by observational studies. It follows that pertussis vaccination failure may be more common, particularly because aP are known to attenuate clinical presentation making vaccinated individuals more difficult to diagnose with pertussis [11,12,13]. AP vaccines likely have a multimodal model of failure (leaky, primary and secondary) and are known to protect best against morbidity and mortality early in life [14]. As such, in the 0–4 age group, the VE of a three-dose aP primary series is highest against pertussis hospitalisation in infants less than one-year-old (93%) and lowest against non-hospitalised notifications in three-year-olds (84%) [9].

Pertussis vaccination failure is the occurrence of pertussis, caused by *B. pertussis* in an individual vaccinated in accordance with the national immunisation schedule [15]. When applied to maternal pertussis vaccination in pregnancy, pertussis vaccination failure is understood as pertussis occurring in her infant within the first six weeks of life (before the infant’s first pertussis vaccination). Vaccination failure is a complex concept and has four broad categories of causes: Vaccinee-related (e.g., immunocompromise), schedule-related (e.g., poor timing and number of doses recommended), vaccine-related (e.g., poor coverage of pathogen strains or serotypes), and usage-related (e.g., cold chain storage failure, suboptimal administration) [15]. Pertussis vaccination failure requires two conditions: Suboptimal vaccine-induced protection and exposure to *B. pertussis*. One risk factor may increase the risk of one or both conditions. Studies of pneumococcal and varicella vaccination failure have begun by investigating known risk factors for pneumococcal and varicella; we share this approach [16,17].

Risk factors for pertussis disease have been outlined previously, although risk factors for pertussis vaccination failure is an area of little prior research [18,19,20,21,22,23,24,25,26]. A number of factors have been associated with susceptibility to *B. pertussis* infection and disease; a brief and nonexclusive list includes low birth weight, prematurity, asthma, immunocompromised conditions and socioeconomic deprivation [18,19,20,21,22,23,24,25,26]. The literature on pertussis vaccination failure is limited and is almost exclusively published from the Niakhar Studies and Senegal Pertussis Trials [27,28,29,30]. The context of these findings (lower middle income country) likely do not fully apply to high-income countries [29,30]. The risk factors reported for pertussis vaccination failure from the Senegal Pertussis Trial included high birth rank, underweight and stunting [29,30].

Vaccine-related reasons for failure are dealt with by extensive ongoing biomedical and pharmaceutical research efforts [31,32,33,34,35]. Therefore, a strategic use of resources is to focus on improving the effectiveness of current aP vaccines through changes to the national immunisation schedule [36]. Limited consideration has been given to biological and social variations in vaccinee circumstances and their interaction with national immunisation schedules that may contribute to vaccination failure. This study pursues an understanding of pertussis vaccination failure in relation to New Zealand′s current national immunisation schedule.

To our knowledge, no studies have investigated pertussis vaccination failure in a high-income country. The main aim of this study is to describe pertussis vaccination failure cases and investigate candidate risk factors for pertussis vaccination failure in the New Zealand context. The identification of risk factors and subsequently higher risk groups is an important step in manipulating the national immunisation schedule for better protection of vulnerable groups; one that aligns with public health goals such as maximising overall health and reducing disparities between groups.

## 2. Materials and Methods

The Health and Disability Ethics Committees (a national-level committee) waived the need for ethics approval (17 January 2017). However, this study received approval from the University of Auckland Human Participants Ethics Committee (UAHPEC) (reference number 018664) on 31 March 2017. Minor amendments to the original UAHPEC application were approved by UAHPEC on 19 December 2017 (reference number 018664).

The research questions for this study are: What evidence is there for risk factors of pertussis vaccination failure?
(a)Between birth and first pertussis immunisation in infants born to mothers who received maternal pertussis vaccination during their pregnancy;(b)in fully vaccinated infants and young children before their four-year pertussis booster;

In order to answer these questions, this study has two components and four sub-studies (see Figure 1).

Study objectives:(1)Describe infants born to mothers Tdap vaccinated during pregnancy with pertussis disease before their first pertussis vaccination, from birth to first pertussis vaccination using Ministry of Health national administrative data;(2)Identify risk factors for pertussis disease in infants of women Tdap vaccinated during pregnancy by determining associations between risk factors identified in Objective 1 and pertussis disease in infants born to women Tdap vaccinated during pregnancy between birth and first pertussis vaccination;(3)Describe fully vaccinated infants and young children with pertussis vaccination failure from birth to four year booster vaccination or four years old using Ministry of Health national administrative data and national notification data;(4)Identify risk factors of pertussis vaccination failure in fully vaccinated infants and young children by determining associations between risk factors identified in Objective 3 and pertussis vaccination failure.

### 2.1. Study Design

The study setting is New Zealand. For each study population, one case series study will be conducted to describe the pertussis vaccination failure cases and one retrospective cohort study will be conducted to identify risk factors for pertussis vaccination failure (Figure 1). The study population characteristics for each objective are described below.

#### 2.1.1. Objective One Inclusion Criteria

All single live birth infants with a gestation ≥28 weeks whose mothers were administered funded pertussis vaccine (Tdap) during pregnancy in New Zealand and with a pertussis notification identified in the national notifiable diseases database or pertussis hospitalisation identified in the national minimum dataset in the period between birth and first infant pertussis vaccination between 1 January 2013 and 31 December 2016 inclusive.

#### 2.1.2. Objective Two Inclusion Criteria

All single live birth infants with a gestation ≥28 weeks whose mothers were administered funded pertussis vaccine (Tdap) during their pregnancy in New Zealand between 1 January 2013 and 31 December 2016 inclusive.

#### 2.1.3. Objective Three Inclusion Criteria

All infants/young children registered in the national immunisation register who meet the study definition of “fully vaccinated” and have a pertussis notification identified in the national notifiable diseases database or pertussis hospitalisation identified in the national minimum dataset occurring between greater than 35 days after the third dose of the primary course and before the four-year pertussis booster dose or four years of age, whichever comes first. These events must have occurred between 1 January 2006 and 31 December 2016 inclusive.

#### 2.1.4. Objective Four Inclusion Criteria

All infants/young children registered in the national immunisation register who meet the study definition of “fully vaccinated” between 1 January 2006 and 31 December 2016. The birth cohort is defined as those born between 1 January 2006 and 31 December 2016, inclusive, using the date of birth from the national health index dataset. The study population will include 11 sub-cohorts, eight of which include children who reached the maximum of four years, nine sub-cohorts reached 2–3 years, 10 reached 1–2 years and 11 reached 0–1 year. Six national administrative datasets were used for all the objectives in this study (see Table 1).

#### 2.1.5. Study Definitions

**Maternal pertussis vaccination:** The administration of a Tdap booster (Boostrix^®^, GlaxoSmithKline, Rixensart, Belgium) to a pregnant woman between 28 and 38 weeks of gestation. The 28–38-week timeframe relates to the current funding guidelines for the maternal pertussis booster in pregnancy vaccination [1]. The New Zealand schedule recommends a maternal pertussis immunisation for every pregnancy [1]. The pregnancy during which the mother was vaccinated is the pregnancy during which the infant in the dataset was carried.

**Fully vaccinated**: The on-time interval-adjusted administration of the primary pertussis course with either DTaP-IPV (Infanrix^®^-IPV, GlaxoSmithKline, Rixensart, Belgium), funded in 2006–2008 for the primary course or DTaP-IPV-HepB/Hib (Infanrix^®^-Hexa, GlaxoSmithKline, Rixensart, Belgium), funded for the primary course since 2008 [36]. This definition involves the integration of principles of immunology [43], guidelines from the Advisory Committee on Immunization Practices [44], and the New Zealand definition of timeliness for the primary pertussis series [45] (see Figure 2). The following conditions must be met to be considered fully vaccinated:(a)First pertussis-containing vaccination administered no earlier than 4 days (inclusive) before 6 weeks of age, and no later than 10 weeks of age;(b)Second pertussis-containing vaccination no later than six weeks after the scheduled age of 3 months;(c)Third pertussis-containing vaccination no later than six weeks after the scheduled age of 6 months;(d)Each pertussis-containing vaccination administered at least 3 weeks apart.

**Maternal pertussis vaccination failure**: One or more pertussis hospitalisation(s) and/or notification(s) occurring between birth and first pertussis vaccination in infants whose mothers met the study criteria for maternal pertussis vaccination. As there is a difference in the level of certainty in the diagnosis, clinically confirmed and clinically suspected cases will be reported separately.
(a)Clinically confirmed maternal pertussis vaccination failure in the infant:The occurrence of pertussis hospitalisation (ICD-AM 10 code A37.0 only) and or notification (“confirmed” status only) in the infant between birth and first pertussis vaccination.(b)Clinically suspected maternal vaccination pertussis vaccination failure in the infant:The occurrence of pertussis hospitalisation (ICD-AM 10 code A37.9 only) and or notification (“probable” and “suspected” status only) in the infant between birth and first pertussis vaccination.

**Primary vaccination series pertussis vaccination failure**: One or more pertussis hospitalisation(s) and/or notification(s) occurring within the follow-up period in fully vaccinated infants/young children. As there is a difference in the level of certainty in the diagnosis, clinically confirmed and clinically suspected cases will be reported separately.
(a)Clinically confirmed primary series pertussis vaccination failure:The occurrence of pertussis hospitalisation (ICD-AM 10 code A37.0 only) and or notification (“confirmed” status only) more than 35 days after the last vaccination in the primary series but before four years of age or the receipt of the four-year pertussis booster vaccination. This timeframe aimed to capture only the individuals considered fully vaccinated at the time of exposure to *B. pertussis* who subsequently developed disease. Thirty-five days allowed 14 days for immune response to the pertussis vaccine and 21 days for the incubation period of pertussis.(b)Clinically suspected primary series pertussis vaccination failure:The occurrence of pertussis hospitalisation (ICD-AM 10 codes A37.8 and A37.9 only) and or notification (“probable” and “suspected” status only) more than 35 days after the last vaccination in the primary series but before four years of age or the receipt of the four-year pertussis booster vaccination.

**Pertussis notification**: any case of pertussis reported to EpiSurv by clinicians or through electronic laboratory alerts with a case classification of ‘suspect’, ‘probable’ and ‘confirmed’ (Table 2). Notifications were limited to non-hospitalised notifications to prevent duplication of the same event and to indicate mild (non-hospitalised notified) versus severe (hospitalised) cases. Notification events were determined using the EpiSurv database, and events must have occurred between 1 January 2006 and 31 December 2016.

**Pertussis hospitalisation**: a principle or other diagnosis on hospital discharge with any of the pertussis related ICD-10-AM codes in Table 3. Hospitalisation events were determined using the national minimum dataset, and events must have occurred between 1 January 2006 and 31 December 2016 inclusive. Both primary and secondary diagnoses for hospitalised cases were included; these will not reported separately.

### 2.2. Multiple Notification and Hospitalisation Events

Multiple pertussis hospitalisations and notifications will be treated as separate events if at least 60 days has lapsed since the preceding pertussis hospitalisation/notification. Sixty days was the estimated length of a pertussis disease event; this allowed for 14 days of incubation (range between 5 and 21 days), 11 days for the catarrhal stage (range between 1 and 2 weeks), 21 days for the paroxysmal stage (range between 1 and 6 weeks) and 14 days for the convalescent stage (range between 1 and 4 weeks) [47].

### 2.3. Statistical Analysis Plan

All data analyses will use SAS statistical software (SAS Institute Inc., Cary, NC, USA). An alpha of 0.05 for statistical significance will be used in all analyses. The frequencies and percentages will be reported for categorical data. Pearson′s chi-square test or Fishers exact test will be used to analyse associations between categorical data. The data will be tested for normally distributed data, *t*-tests will be used to test the differences between the means for normally distributed data, Mann–Whitney or Kruskal–Wallis tests will be used for non-normally distributed data. The continuous data will be reported as means and standard deviations, unless there are more appropriate statistics to report (for example, median). A pattern analysis will be used to investigate possible risk factor patterns in the cohort study. The variables included in this analysis will be those deemed relevant as per existing literature and case series study findings. The thresholds will be based on the distribution of the data or other relevant published information (epidemiological, clinical and statistical). As part of Objective 1, the number and the percentage of infants born to women who did not receive pertussis immunisation during their pregnancy will be reported.

Appropriate statistical modelling methods will be used to test the predictive power of the potential risk factors for pertussis. A survival analysis where the first event was pertussis diagnosis will be used. The covariates will include age, sex, ethnicity, deprivation and geographical area. The additional covariates for Objective 2 will include variables that influence the efficacy of maternal/infant antibody transfer e.g., time between maternal vaccination and delivery and number of previous maternal pertussis pregnancy vaccinations.

#### Power Statement for Objective 4

As the sample size is fixed, the power available to detect a statistically significant difference (α = 0.05) between the exposure groups was calculated using OpenEpi [48]. As the exposure variable selection had not occurred yet, socioeconomic deprivation was chosen as an example for power calculation because there is a known association between socioeconomic deprivation (measured by the New Zealand Deprivation Index 2013 (NZDep13)) and infectious respiratory diseases in New Zealand [49,50]. Hospitalised and notified cases are unlikely to be homogenous and as the hospitalised cases are likely the smallest group (approximate *n* = 85), the hospitalised cases are the basis for the power calculation. The sample size between the 2006 and 2016 study period is estimated to be 504,984 infants and young children.

There was an approximate 40% exposure to socioeconomic deprivation (NZDep13 deciles 7–10) and a 0.02% rate of pertussis vaccination failure events during the follow up period in the deprived group and 0.01% in the not-deprived group. It was estimated that this study had 84% power to detect a difference for Objective 4 at a significance level of 0.05% (Table 4).

## 3. Discussion

Pertussis vaccination failure research is pauce with critical knowledge gaps. One such gap is whether there is evidence for high-risk groups for pertussis vaccination failure (beyond the immunocompromised). High-risk groups are significant for the pursuit of equity in public health and for strategic interventions for disease control and prevention. This study aims to improve the use of the currently available aP vaccine by investigating ‘who’ may be at greater risk for pertussis vaccination failure under the current primary immunisation schedule in New Zealand. A better understanding of the relationship between risk factors and pertussis occurring between birth and first pertussis immunisation in infants born to mothers pertussis immunised during pregnancy may also help to develop pertussis prevention strategies for this group. Finally, estimating how common pertussis vaccination failure is in infants and young children and how common pertussis is between birth and first pertussis vaccination are important epidemiological information required for better pertussis control.

The use of large linked administrative datasets offers significant advantages, such as enabling the study of rare events like pertussis vaccination failure and detailed, verifiable and longitudinal information. The use of national administrative data also limits the choice of variables to that which is available and of a reasonable quality. Some study limitations will therefore be related to the validity of the variables chosen for study purposes and the quality and reliability of these variables over time. The case series studies are exploratory in nature, however, it is a valuable and necessary starting point for further research. A comprehensive description of demographic, clinical and pharmaceutical variables in relation to pertussis vaccination failure has not been published before. The complimentary use of exploratory and hypothesis testing study designs is a significant strength of this study, as this design minimises potential issues with the case series studies and maximises the potential of the cohort studies by making available high-quality information based on more than just investigative hunches. This study will begin to address a major gap in paediatric pertussis prevention literature.

## Figures and Tables

**Figure 1 vaccines-07-00065-f001:**
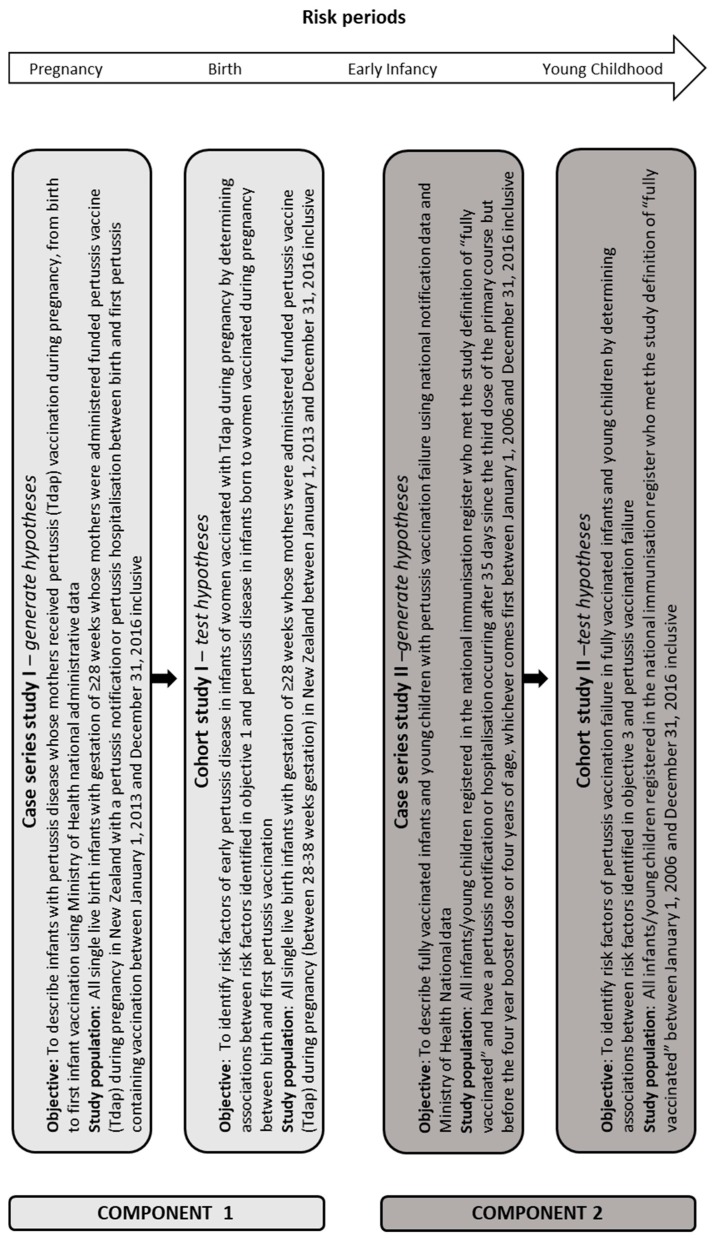
Study design and study objectives.

**Figure 2 vaccines-07-00065-f002:**
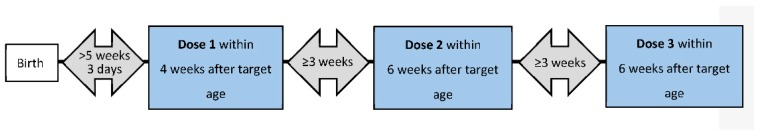
Fully vaccinated criteria.

**Table 1 vaccines-07-00065-t001:** Data sources.

Data Source	Information	Relevant Fields
National Health Index database	The national health index (NHI) database holds static demographic information such as the NHI number, sex, date of birth, socioeconomic deprivation and ethnicity [37]. Some demographic information such as address, level of socioeconomic deprivation and ethnicity are changeable and therefore amenable to regular updates in real-time (at every presentation to health care services). The NHI number is a unique randomly assigned alphanumeric identifier that enables accurate identification of individuals for medical care and administrative records in New Zealand. The NHI number links all patient information from national and regional health services [37].	Encrypted NHI number; Date of birth; Sex; New Zealand Deprivation Index 2013 (NZDep13) decile; Ethnicity
National Immunisation Register	An electronic collection of all registered immunisation enrolments and events of children in New Zealand and from 2005, these are updated weekly [38]. This collection does not necessarily contain all infants and young children who were vaccinated, parents or legal guardians can withdraw their child′s information from the national immunisation register collection.	Encrypted NHI number; Vaccine type; Antigen type; Vaccination date; Vaccination status;
National Minimum Dataset	A health statistic collection dataset containing clinical and other information about public and private hospital discharges for inpatients and day patients [39].	Encrypted NHI; Admission and discharge date; Length of stay; ICD-10 diagnosis code(s)
Maternity collections	A collection of the demographic and clinical features of women in New Zealand using publicly funded maternity/new-born services from 9 months before birth to 3 months after [40].	Encrypted NHI; Birth weight; Gestational age; Other inpatient and day-patient event data across pregnancy, birth and postnatal periods pertaining to both fetus/infant and mother
Pharmaceutical collection	A system used for management of subsidised pharmaceuticals in New Zealand [41].	Encrypted NHI; Anatomical Class; Chemical name; Duration of supplied pharmaceuticals
National Notifiable Diseases Database (EpiSurv)	EpiSurv is a web assessable application operated by the Institute of Environmental Science and Research built using the Surveillance information New Zealand framework [42]. EpiSurv holds the national notifiable disease database and allows the real time recording of notifiable disease cases across New Zealand from public health services in New Zealand. EpiSurv is under contract from the Ministry of Health to collect demographic, clinical and risk factor information of reported cases.	Encrypted NHI; Disease; Notification status; Risk factors for disease; Report date; Onset date;

**Table 2 vaccines-07-00065-t002:** Environmental Sciences and Research Limited Notification Classification System and criteria [46].

Classification	Definition
Suspect	Idiopathic presentation of any paroxysmal cough with whoop, vomit or apnoea
Probable	Presentation clinically compatible with pertussis with a high *B. pertussis* IgA test or a significant (fourfold increase in titres) in antibody levels between paired sera at the same laboratory
Confirmed	Clinically compatible presentation with either laboratory confirmed pertussis infection (*B. pertussis* only) or epidemiologically linked to a confirmed case

**Table 3 vaccines-07-00065-t003:** ICD10 AM diagnostic codes for pertussis.

ICD10-AM Code	Description
A37.0	Whooping cough due to *B. pertussis*
A37.8	Whooping cough due to *Bordetella* species
A37.9	Whooping cough, unspecified

**Table 4 vaccines-07-00065-t004:** Power calculation.

	Exposed (Deprived, NZDep13 Deciles 7–10)	Not-Exposed (Not Deprived, NZDep13 Deciles 1–6)
Sample size	219,814	285,800
Risk of disease	53 cases ÷ 219,814 × 100 = 0.02%	32 cases ÷ 285,800 × 100 = 0.01%

Power based on normal approximation without continuity correction = 84% [48].

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
