# Peer review of "Pertussis Vaccination Failure in the New Zealand Pediatric Population: Study Protocol"

_vaccines, 2019, doi:10.3390/vaccines7030065_

Round 1

Reviewer 1 Report

The study protocol is important, clear, and well-developed.  The only suggestion I have is to include in the study the number and percentage of infants with pertussis who were born to women who did not receive the DTap during pregnancy.  This statistic will show the importance of administering the vaccine to pregnant women by revealing how many cases of pertussis are prevented.

Author Response

We thank the reviewer for the time taken to review our manuscript and their constructive comments. We agree that although reporting maternal immunisation coverage is not a study aim it is indeed important to provide this statistic for context. We have included this in the protocol under the section ‘statistical analysis plan’.

Reviewer 2 Report

The presented manuscript addresses a topic with impact on public health, such as the pertussis vaccine failure in the pediatric population. In particular, the authors present a study protocol.
The topic is relevant and the auhtors adequately designed the protocol;

Below I include a few comments for a better interpretation

1- In the introduction, it would be important that the authors expand the information on their statement on the effectiveness of aP against clinical pertussis disease after the primary vaccination series (lines 40-41). Type of study, type of vaccine, etc

2-- Regarding maternal immunization, does the recommendation in New Zealand stand for all pregnancies? or for a single pregnancy?

If it is valid for all pregnancies, it would be important to discriminate number of maternal doses when making the analyzes on failure and risks.

3- It would be important to register the brand and the composition of the vaccines to be used both in the pregnancy  and  the primary series to then perform a differential analysis taking into account the brands.

Minor comments

in B. pertussis please introduce an space between B. and pertussis

line 134 please include the reference

Author Response

We thank the reviewer for the time taken to review our manuscript in detail and for their constructive comments. 

1 – We have taken into consideration the valid comment made and have altered the statement to more accurately summarise important aspects of aP VE (now lines 40 - 56).

2 – the recommendation in New Zealand stands for each pregnancy. We agree that previous maternal pertussis immunisation in pregnancy is an important covariate and will be included in the analysis. We have included this in the protocol in the section ‘statistical analysis plan’.

3- We agree it is important to register the brand and compositions of the vaccines used, we have amended the protocol to include this information. Only one brand GlaxoSmithKline has been used for both pregnancy vaccinations and primary series vaccinations so a differential analysis is not possible. Specifically, only Boostrix has been used for pregnancy immunisation and Infanrix-IPV (2006 and 2007) and Infranrix-hexa (2008 and 2016) for the primary series.

4. The line 134 error referenced a table that was removed from the manuscript before submission, the error message has been removed from the manuscript. 

Reviewer 3 Report

This is a very worthy topic of inquiry and I wish the authors the best as they work on their study. I have some minor comments to help guide their writing.

The abstract is a bit confusing. The authors mention two aims, a case series design and a retrospective cohort study, as well as five datasets. I would like these more clearly linked – which aim goes with what study design and dataset. And what are you linking together with a unique identifier across datasets? (potentially not even necessary to mention linkages in abstract and could simply say “National administrative datasets will be used to ….” (Figure 1 is relatively clear in this respect – I think you just need to reformulate it into abstract)

-        also Table 1 lists six datasets? (vs five mentioned in abstract and line 135)

Minor copy editing issue – but I think sometimes you don’t have spaces in between period and next sentence (lines 35, 36, etc.)

Introduction line 41: “vaccination is known….” is this in reference to wP or aP or both?

Line 83: What is the Health and Disability Ethics Committee? Is this a national level thing or a university level board?

Line 88: Research question**s**?

I’m not sure line 206 (Statistical significance will be interpreted using statistically significance effect sizes, confidence intervals, sample size, and study design) is that necessary (and in particularly not sure what sample size and study design are doing in this list)

Will you have access to birth mothers’ vaccination records from before the pregnancy? And will you know if they would have received wP or aP?

Author Response

We thank the reviewer for the time taken to review our manuscript in detail and for their constructive comments and writing guidance.  

1. We have re-written the abstract so that the methods, objectives and study components are clearer. We have also remedied the incorrect number of data sets cited, there are indeed 6.

2. We have clarified this statement (now lines 51 - 52) to "aP are known to attenuate clinical presentation and vaccinated individuals less likely to be correctly diagnosed with pertussis" it is also preceeded by a more suitable discussion on aP.

3. Apologies, the Health and Disability Ethics Committees are ‘Ministerial committees’, at the national level. We have clarified this in the manuscript.

4. We have removed this sentence (line 206).

5. Unfortunately we cannot determine birth mothers’ immunisations prior to pregnancy because our data source (National Immunisation Register) captures childhood immunisations from those born from 2006 and other selected programme immunisations including pregnancy (from 2013). However in New Zealand aP replaced wP in 2000 – making it likely that most of the birth mothers were wP vaccinated or not vaccinated at all. We realise not being able to establish mothers vaccination history with certainty is a limitation, particularly for the study looking and maternal immunisation failure in infants less than 6 weeks old as there is a documented greater vaccine effectiveness with whole cell priming.